# A Novel Cooperative Path Planning for Multi-robot Persistent Coverage with Obstacles and Coverage Period Constraints

**DOI:** 10.3390/s19091994

**Published:** 2019-04-28

**Authors:** Guibin Sun, Rui Zhou, Bin Di, Zhuoning Dong, Yingxun Wang

**Affiliations:** 1School of Automation Science and Electrical Engineering, Beihang University, Beijing 100083, China; zhr@buaa.edu.cn (R.Z.); dongzhuoning@buaa.edu.cn (Z.D.); wangyx@buaa.edu.cn (Y.W.); 2National Innovation Institute of Defense Technology, Academy of Military Sciences PLA China, Beijing 100071, China; dibin@asee.buaa.edu.cn

**Keywords:** multi-robot, cooperative coverage, persistent coverage, path planning, coverage period constraints, obstacle avoidance

## Abstract

In this paper, a multi-robot persistent coverage of the region of interest is considered, where persistent coverage and cooperative coverage are addressed simultaneously. Previous works have mainly concentrated on the paths that allow for repeated coverage, but ignored the coverage period requirements of each sub-region. In contrast, this paper presents a combinatorial approach for path planning, which aims to cover mission domains with different task periods while guaranteeing both obstacle avoidance and minimizing the number of robots used. The algorithm first deploys the sensors in the region to satisfy coverage requirements with minimum cost. Then it solves the travelling salesman problem to obtain the frame of the closed path. Finally, the approach partitions the closed path into the fewest segments under the coverage period constraints, and it generates the closed route for each robot on the basis of portioned segments of the closed path. Therefore, each robot can circumnavigate one closed route to cover the different task areas completely and persistently. The numerical simulations show that the proposed approach is feasible to implement the cooperative coverage in consideration of obstacles and coverage period constraints, and the number of robots used is also minimized.

## 1. Introduction

Research interest for the coverage and coordination of multi-agent has shown an increase in the field of artificial intelligence (AI) and control [1,2,3,4,5,6]. In particular, coverage control using multiple robots with limited sensing capabilities has received significant attention recently due to its versatility in many applications, such as mapping, patrolling, surveillance, and complete coverage [3,4,5,6,7,8,9,10,11]. However, it is more difficult to achieve persistent coverage for a group of multiple robots considering obstacle avoidance and coverage period, because mission domains may have differently shaped obstacles, as well as more complicated constraints.

In the current literature, many advanced methods, such as grid-based coverage, cellular decomposition and topological coverage, have been proposed for coverage problems [12,13]. The literature [13] develops two efficient coverage strategies for multiple robots based on boustrophedon cellular decomposition to achieve complete coverage of a known environment. Traditional AI search algorithms, such as A-Star and its variants, have also been applied [14], but cannot adapt to multiple robots. Votion and Cao first develop three improved A-star algorithms to obtain the optimal path, and then they present a new spatially diverse path planning algorithm based on the A-star variants to address the need for path diversity in multi-agent path planning [15]. Spanning tree coverage [16] is developed for multi-robot area patrolling [17] and surveillance [18], but ignores the sub-regions with different importance. Market-based mechanisms have also been used to assign work to robots, but their applications have been limited [8]. At the other end of the spectrum is the coordination of multi-agent, which includes potential-function based approaches, digital pheromone mechanisms, and particle swarm optimization (PSO) [19,20,21,22]. They are effective in dealing with the problem, but often suffer from troubles of local optimum. Another coverage approach includes artificial neural networks (ANNs) to represent control politics [23]. Multi-robot persistent coverage of the convex polygonal region is investigated in [24], where the path consists of the vertices of the scaled barycentric polygons. In [25], an adaptive path planning algorithm is proposed for multiple AUVs cooperative environmental sampling and sensing over an interest region. Atınç, Stipanović, and Voulgaris study a dynamic coverage for multi-agent systems, where the main objective of a group of mobile agents is to explore a given compact region [26]. Franco et al. present a new bounded potential repulsion law to achieve persistent coverage for a team of agents with collision avoidance [19]. The speed controller along the path is further developed in [27] for persistent awareness coverage using mobile sensors. The robots are supposed to be placed on the path uniformly to increase the frequency of revisits in [17]. It is also assumed that the coverage periods of the areas in the region are equal in the literature [27]. Persistent monitoring of given discrete sites under different frequency constraints is considered in [28,29], where the travelling salesman problem (TSP) [28] or the vehicle routing problem with time windows [29] is solved for the routing of the robots.

Most of these methods for multi-robot persistent coverage can be classified into two categories. One class is focussed on the paths to cover the task areas completely and persistently [10,30,31]. They present some new path planning algorithms and implementation for the efficient complete coverage of a known area. However, these methods are only suitable for simple task environments. In addition, they also ignore the sub-regions in the mission domain, which may be more important and need to be re-covered more frequently. The other class is characterized by approaches which are simple and highly scalable to address the problem of multi-robot persistent coverage [32,33]. These approaches can have better performance than a single robot for persistent coverage, but they do not consider using the least number of the robots to cover the areas.

Although the aforementioned methods have promoted the development of coordination algorithm for multi-robot persistent coverage, the number of robots used and coverage period are not taken into account in these cooperative persistent coverage methods. Therefore, this paper presents a new combinatorial approach for cooperative multi-robot path planning, which focuses on the persistent coverage problem with obstacles and different coverage period constraints using the minimum number of robots. The robots are supposed to re-cover the region of interest within every unit of time periodically. The contribution of the combinatorial method with respect to previous works is summarized as three-fold. The first contribution is the development of the proposed path planning based on sensor deployment for cooperative persistent coverage in complex task environments with obstacles. Compared with the traditional coverage planning, the new approach divides path planning issues into sensor deployment problem (SDP) and TSP in order to effectively cope with the planning puzzle caused by environmental obstacles. More precisely, the proposed algorithm introduces the idea of sensor deployment to implement coverage planning in more complex environments, thereby improving the adaptability and robustness of the algorithm to the environment. The second contribution is to consider the coverage period constraints of different sub-regions in the mission domains. The coverage period is utilized to indicate how frequently the region should be re-covered. In the mission area, there may be some sub-regions of different importance depending on the target probability density. Some sub-regions of greater importance are required to be re-covered with smaller periods. In addition, the sensors are deployed to cover the sub-region with the smallest coverage period first, then the sub-region with larger coverage period next, and so on. The third contribution is to optimize the number of robots performing the task for purpose of using a minimum number of robots to achieve persistent coverage for the mission area. Additionally, the approach can adaptively adjust the number of robots used according to the coverage period constraints of different sub-regions. 

The rest of this paper is organized as follows. Section 2 presents the preliminaries to the improved cooperatively coevolving particle swarm optimization (CCPSO2) and modified genetic algorithm (GA). Section 3 describes the problem formulation of multi-robot persistent coverage. In Section 4, a new combinatorial approach for cooperative path planning is developed to achieve the multi-robot persistent coverage. The numerical results of proposed approach are discussed in Section 5. Finally, the discussion and conclusions are made in Section 6 and Section 7, respectively.

## 2. Preliminaries

In this paper, we use the CCPSO2 because it is fast enough to find the optimal solution of SDP. In addition, modified GA is utilized to solve TSP because of its inherent parallelism and global search capability.

### 2.1. Improved Cooperatively Coevolving Particle Swarm Optimization

PSO is one of popular swarm intelligence methods. However, the performance of the PSO algorithm deteriorates rapidly as the particle dimension increases. In contrast, the CCPSO2 algorithm decomposes the solution vector into different parts and each part is optimized with a single particle swarm. It reduces the dimension of the solution vector in a single particle swarm and has advantages in solving large-scale optimization problems. Therefore, in consideration of SDP characteristics, CCPSO2 with multiple heuristic rules is introduced to enhance particle diversity and algorithm performance.

In the PSO algorithm, each particle represents a potential solution to the optimization problem. Particles move through the search space to seek the best position. Assume that $x_{i}$ and $y_{i}$ are the current position and local optimal position of ith particle respectively. Let $\hat{y}$ be the global optimal position of particles in the swarm. The update of $y_{i}$ and $\hat{y}$ are determined by:(1)$$\left\{ \begin{array}{l} y_{i}=x_{i},\;\mathrm{if}\;f(x_{i})<f(y_{i})\\ \hat{y}=\arg\min(f(y_{i})) \end{array}\right.$$
where $f(x_{i})$ refers to the fitness value of the particle.

The update of velocity and position are formulated as [34]:(2)$$\left\{ \begin{array}{l} v_{i,d}(t+1)=\omega (t)v_{i,d}(t)+c_{1}r_{1}(t)(y_{i,d}(t)-x_{i,d}(t))+c_{2}r_{2}(t)({\hat{y}}_{i,d}(t)-x_{i,d}(t))\\ x_{i,d}(t+1)=x_{i,d}(t)+v_{i,d}(t+1) \end{array}\right.$$
where $v_{i,d}(t)$ and $x_{i,d}(t)$ are the velocity and position of the ith particle in the dth dimension respectively; ω(t) is the inertial weight; $c_{1}$ and $c_{2}$ are acceleration constants; $r_{1}(t)$ and $r_{2}(t)$ are random numbers and satisfy $r_{1}(t)\in [0,1]$, $r_{2}(t)\in [0,1]$.

In order to improve the particle diversity and prevent premature convergence to local optimum, a new update model that uses both Gaussian and Cauchy distributions, as well as ring topology, is proposed in [35].
(3)$$x_{i,d}(t+1)=\left\{ \begin{array}{l} y_{i,d}(t)+C(1)\left|y_{i,d}(t)-{y^{\prime }}_{i,d}(t)\right|,\;\mathrm{if}\;rand<p\\ {y^{\prime }}_{i,d}(t)+N(0,1)\left|y_{i,d}(t)-{y^{\prime }}_{i,d}(t)\right|,\;\mathrm{otherwise} \end{array}\right.$$
where C(1) and N(0,1) are the random numbers generated following the Cauchy and Gaussian distributions respectively; rand is a random number generated uniformly in the range of [0,1]; p is a custom parameter for Cauchy sampling to occur; ${y^{\prime }}_{i}$ denotes a local neighborhood best for the ith particle.

In this paper, the ring topology is utilized to describe the particles’ neighborhood. Each particle is supposed to have an immediate left and right neighbor. Therefore, ${y^{\prime }}_{i}$ can be defined as:(4)$${y^{\prime }}_{i}=\underset{y_{i}}{\arg\min}(f(y_{i-m}),\cdots ,f(y_{i}),\cdots ,f(y_{i+m}))$$
where m is the neighborhood range. It increases with the number of iterations in this paper. A small m can improve the particle diversity at the beginning of the optimization process and a larger m could benefit the convergence at the latter stage of the optimization.

The n-dimensional solution vectors are decomposed into K components in the CCPSO2 algorithm. Each component corresponds to a swarm with s dimensions, where s=n/K. Suppose that $P_{j}.x_{i}$ and $P_{j}.y_{i}$ are the current position and the local optimal position of the ith particle of the jth swarm. Let $P_{j}.\hat{y}$ be the global optimal position of the jth swarm. The current best context vector can be given by $\hat{y}=(P_{1}.\hat{y},P_{2}.\hat{y},\cdots ,P_{s}.\hat{y})$.

In order to evaluate the ith particle of the jth swarm, substitute $P_{j}.x_{i}$ for $P_{j}.\hat{y}$ in $\hat{y}$. Hence, define the following combination of particles:(5)$$b(j,P_{j}.x_{i})=(P_{1}.\hat{y},P_{2}.\hat{y},\cdots ,P_{j-1}.\hat{y},P_{j}.x_{i},P_{j+1}.\hat{y},\cdots ,P_{s}.\hat{y})$$

The $P_{j}.y_{i}$ can be updated as follows:(6)$$\left\{ \begin{array}{l} P_{j}.y_{i}=P_{j}.x_{i},\;\mathrm{if}\;f(b(j,P_{j}.x_{i}))<f(b(j,P_{j}.y_{i}))\\ P_{j}.\hat{y}\;=P_{j}.y_{i},\;\mathrm{if}\;f(b(j,P_{j}.y_{i}))<f(b(j,P_{j}.\hat{y})) \end{array}\right.$$

Considering the properties of SDP, several heuristic operators for the update of the particles’ positions are introduced to improve the convergence speed of the CCPSO2 algorithm.
Addition. If there are uncovered cells and sensors that have not been deployed in the particle, a sensor is randomly chosen to place near one uncovered cell.
(7)$$\mathrm{if}\;c_{i}=0,\;\mathrm{then}\;\mathrm{add}\;s_{n+1}\;\mathrm{at}\;\mathrm{point}\;c_{i}$$
where $c_{i}$ is a binary vector representing the ith discretized cell (for details, refer to Section 4.2); $s_{n+1}$ indicates the new sensor added to the existing n sensors.Movement. If there are uncovered cells in the region and all sensors in the particle have been deployed, a sensor is chosen to move a short distance towards one uncovered cell.
(8)$$\mathrm{if}\;c_{i}=0,\;\mathrm{then}\;\mathrm{move}\;s^{\prime }\;\mathrm{to}\;\mathrm{point}\;c_{i}$$
where $s^{\prime }$ denotes a certain sensor around the cell $c_{i}$.Deletion. If the distance between any deployed sensor and other sensors in the global best is less than the sensing radius of the sensors, the deployed sensor in the particle is deleted.
(9)$$\mathrm{if}\;\left\|p(s_{i})-p(s_{j,j\neq i})\right\|<r_{s},\;\mathrm{then}\;\mathrm{delete}\;s_{i}$$
where $p(s_{i})$ is the position of the ith sensor and ‖·‖ refers to the Euclidean distance between $p(s_{i})$ and $p(s_{j})$.Fusion. After all the cells of region have been covered, the fusion operation can be performed. If the distance between any two deployed sensors is less than a certain constant, the two sensors are fused into one sensor with its position at the middle of the two sensors.
(10)$$\mathrm{if}\;\left\|p(s_{i})-p(s_{j,j\neq i})\right\|<\tau ,\;\mathrm{fusion}\;s_{i}\;\mathrm{and}\;s_{j}$$
where τ represent a certain constant.

### 2.2. Modified Genetic Algorithm

GA is a kind of random search method that has evolved from the evolutionary laws of the biosphere. It could use the probabilistic optimization method to adjust search direction adaptively in the process of solving combinatorial optimization problem. GA is one of the most ideal approaches in solving the TSP because of its inherent parallelism and global search capability. Therefore, the TSP is solved by GA to obtain the closed path in this paper.

Assume that there are n cities and each city is represented by an integer from 1 to n. In this way, chromosome $R_{k}$ can be described as $R_{k}=\{g_{1},g_{2},g_{3},\cdots ,g_{n}\},\;g_{i}\in [1,n]$, where $g_{i}$ is the ith gene and also represents a city. $R_{k}$ is a chromosome and refers to the kth feasible path as well. It is a gene sequence consisting of n genes and each gene of the chromosome is different from each other. For example, there are 5 cities in the TSP, then {1,3,4,2,5} is a legitimate chromosome and a possible optimal solution. It means that the salesman visits cities 1, 3, 4, 2, and 5 in order. Suppose that X(j) represents the jth population. It can be expressed as $X(j)=\{x_{1}^{j},x_{2}^{j},x_{3}^{j},\cdots ,x_{m}^{j}\}$, where m is the size of the population; $x_{i}^{j}$ refers to the ith individual of the jth population. In this paper, a random function is used to generate an initialization population $X(0)=\{x_{1}^{0},x_{2}^{0},x_{3}^{0},\cdots ,x_{m}^{0}\}$.

There are three basic operations in the standard GA, that is, selection operation, crossover operation, and mutation operation. Considering the characteristics of the TSP, this paper uses following four operations: selection operation, mutation operation, evolutionary reversal operation, and slide operation.

The selection operation is to generate a new population with higher fitness value. It is selected by the selection probability from the current population. The main objective of selection operation is to inherit the high-quality genes to the next generation while ensuring fast global convergence. In this paper, the individual with the maximum fitness value $f(x_{i}^{j})$ is utilized to generate a new population as the elite individual.

Mutation operation is a very important operator of the GA. In this paper, the mutation operation adopts a strategy of randomly exchanging two genes of one chromosome. As depicted in Figure 1a, there is a chromosome R consisting of 7 genes and randomly generate two gene positions $z_{1}=2$ and $z_{2}=6$. Then exchange the genes at these two positions to obtain the new chromosome $R^{\prime }$. 

In order to improve the local search ability of the GA, the evolutionary reversal operation is introduced after the selection operation and mutation operation. The evolution refers to the unidirectionality of reversal operator, that is, after the reversal operation, the individual will perform the operation if it becomes better, otherwise the reversal is invalid. The method is to randomly generate two random numbers $z_{1}$ and $z_{2}$, and then the genes between $z_{1}$ and $z_{2}$ are re-sorted in reverse order. The process of the evolutionary reversal operation is depicted in Figure 1b. Assume that there are 7 genes in one chromosome and randomly generate two gene positions $z_{1}=3$ and $z_{2}=6$. Then re-sort the genes between $z_{1}$ and $z_{2}$ in reverse order.

Slide operation can greatly inherit the advantages of the parent individual and it can also prevent the algorithm from falling into local optimum. Figure 1c shows the process of slide operation. Firstly, two gene positions $z_{1}=2$ and $z_{2}=5$ are randomly generated. Then rotate one gene position left between $z_{1}$ and $z_{2}$.

## 3. Problem Formulation

### 3.1. Basic Assumptions

In this paper, a new combinatorial approach is proposed for multi-robot persistent coverage. The following conditions are assumed to describe the cooperative persistent coverage.
Each robot is supposed to be homogeneous.The detection zone of a sensor is simplified to a circle, and the influence of robot motion on sensing range is not considered.The center of the sensor detection zone, namely the center of detection circle, is considered as the position of the sensor.The position of a sensor is regarded as the robot observation point, that is, the robot waypoint. When the robot is at the observation point, the area within the coverage of the sensor can be detected.The probability density of target in each sub-region is known by the priori information, and there are different coverage period requirements for each sub-region.The total velocity of each robot is set to the constant value.

### 3.2. Problem Statement

Suppose that there is a group of robots performing persistent coverage task over the given region. The sensing radius of each robot is denoted as $r_{s}$. The region of interest $R_{OI}$ consists of the feasible region $R_{f}$ and the obstacle region $R_{o}$, which satisfy $R_{f}\cup R_{o}=R_{OI}$ and $R_{f}\cap R_{o}=\emptyset$. The obstacles cannot be reached and would also block the sight of the sensors.

The robots are tasked to re-cover the feasible region within every T units of time in order to update their observations continually. The coverage period T is adopted to indicate how frequently the region should be re-covered. Some sub-regions of more importance in the feasible region may be required to be re-covered with smaller periods. It is assumed that the coverage periods of the feasible region and the sub-regions are known according to the prior information. As depicted in Figure 2, there are three sub-regions (sub-regions 1, 2, and 3) and two obstacles (obstacles 1 and 2) in the given region. Sub-regions 1 and 2 are more important and required to be re-covered with smaller coverage periods $T_{1}$ and $T_{2}$, respectively. The rest of the feasible region is denoted as the sub-region 3 with coverage period $T_{3}$. 

Given the capabilities of the robots and the coverage period of each sub-region, the persistent coverage problem is translated into how to plan the robots’ paths with the coverage periods satisfied, using a minimum number of robots. There are two correlative questions in the persistent coverage problem: (1) How to plan a closed path that can effectively avoid geometrical obstacles; (2) How many robots are required at least to satisfy the coverage period constraints? 

In this paper, we first consider the path planning for complete coverage in the region. Due to the complexity caused by the obstacles, the path planning problem is decomposed into SDP and TSP. Then, partition the path into several segments considering the sub-regions with different task importance and the robots with optimum quantity. In order to solve the problem efficiently, we divide the solution strategy into three steps. Firstly, the feasible region $R_{f}$ is covered completely using the virtual sensors with sensing radius $r_{v}$. The purpose of SDP research is to deploy the sensors in the region to satisfy coverage requirements and ensure minimum cost [36,37,38,39,40,41,42]. Sensor deployment in the feasible region can be performed in several stages, according to the coverage periods of sub-regions from small to large. The sub-region with the smallest coverage period is preferentially deployed. Then the sub-region with the larger coverage period is deployed until all the sub-regions are completely covered. Secondly, taking the positions of the deployed sensors as the waypoints, the TSP is solved to obtain the closed path which connects all the sensors. The feasible region can be covered completely if a robot circumnavigates the closed path once. Thus, we call the closed path ‘the frame’. At last, in consideration of the coverage period constraints and the optimum quantity of robots, the frame is partitioned into several segments with the aim of minimizing the number of the segments. On the basis of each partitioned segment, the closed route is generated for each robot. Furthermore, each robot is assigned one closed route to circumnavigate in order to achieve persistent coverage of the feasible region.

The purpose of this paper is to develop a new combinatorial approach for multi-robot persistent coverage, which aims to cover mission domains with different task periods while guaranteeing both the obstacle avoidance and the optimum number of robots. In this paper, the improved CCPSO2, modified GA, and PSO are applied to design the combinatorial algorithm.

## 4. Combinatorial Approach for Multi-Robot Persistent Coverage

### 4.1. Traditional Cooperative Persistent Coverage

Multi-robot cooperative coverage can effectively improve mission efficiency in the task of persistent coverage. In a relatively simple environment, in order to ensure complete coverage of the area, the geometry-based approach is often used to search the area. Figure 3a illustrates an example of the parallel scanning coverage. After determining the number of robots, the routes of each robot are generated according to the geometric rules, in consideration of sensor’s detection width, turning radius, and entering direction. Under this scan strategy, the increase in the number of robots widens the overall scan radius, which reduces the search period and improves coverage efficiency. Sequential scanning coverage is shown in Figure 3b. The method first obtains the path of a single robot persistent coverage. Then each robot moves on the planned path sequentially. Robots are evenly distributed on the route at a certain interval, which reduces the coverage period and achieves a better coverage. The geometry-based approach is a common strategy for multi-robot coverage. However, this approach is only suitable for the regular task area with no obstacles. It cannot effectively address persistent coverage issues in complex task environments with target probabilities and obstacles.

Area decomposition technique is another strategy for multi-robot persistent coverage. The approach first divides the mission area into several sub-regions, and then the robots are tasked to re-cover the respective sub-regions. The literature [43] shows a cooperative search using multiple unmanned air vehicles (UAV). The algorithm divides mission area into several sub-regions in terms of the UAVs’ initial positions and the percent of search area of each UAV. Then each UAV searches respective sub-region and selects the appropriate direction to reduce the number of turns. The planning result of Ref. [43] is illustrated in Figure 3c. The advantage of area decomposition is that the straight part of the coverage route is longer, which can reduce the number of turns and boost the coverage efficiency. However, it is necessary to consider the influence of obstacles on the area division when the mission environment becomes more complicated. At this time, it is extremely difficult to divide the region according to geometric rules. In addition, methods based on area decomposition neglect the sub-regions in the mission area, which may have different importance and need to be re-covered with different coverage periods. 

In summary, current methods for multi-robot persistent coverage mainly have the following defects: 

**Problem 1**. They have failed to address cooperative persistent coverage effectively in complex mission environments with obstacles.

**Problem 2**. Previous works neglect the coverage period of each sub-region in the mission area. Some sub-regions of more importance may be re-covered with different coverage periods.

**Problem 3**. Current methods do not consider how to divide the task region and assign the robots, in order to improve coverage efficiency.

Based on the above facts, the combinatorial method proposed in this paper focuses on the multi-robot persistent coverage with obstacles and different coverage period constraints, using a minimum number of robots. Cooperative persistent coverage can be divided into the following three steps to deal with: (1) sensor deployment in the task area; (2) path planning based on the TSP; (3) partition of closed path considering coverage period.

### 4.2. Sensor Deployment in The Task Area

The coverage region is a bordered area. Once the mission area is given, it can be divided into many rectangular cells whose side length $l_{c}$ is much smaller than the sensing radius $r_{s}$. Let $C_{f}=\{c_{1},c_{1},\cdots ,c_{m}\}$ be the set of cells in the feasible region $R_{f}$. As shown in Figure 4a, the task region is divided into rectangular cells of 50×50 and the side length of each cell satisfies $l_{c}\ll r_{s}$. 

The feasible region is covered completely using the virtual sensors with sensing radius $r_{v}$, where $r_{v}=2r_{s}$. Suppose that $S=\{s_{1},s_{2},\cdots ,s_{n}\}$ is the set of the sensors. Figure 5 illustrates an example of sensor deployment. The task area is covered by the virtual sensors. The blue line is a segment of the frame, namely the closed path connecting all the waypoints in the mission domains. The red closed route is generated on the basis of portioned segment and robot’s sensing range. The robot carrying detection sensor, with sensing radius $r_{s}$, is tasked to track the red route. As depicted in Figure 9, one sensor covers two-part routes of the red closed route. Therefore, the radius of virtual sensor is twice the sensing radius of robot, that is $r_{v}=2r_{s}$.

The cell $c_{i}$ can be covered by the sensor $s_{j}$ if all the four vertices of the cell $c_{i}$ are within the sensing range of the sensor $s_{j}$ and are not blocked by any obstacle. The coverage situation of the sensors is shown in Figure 4b. Let $e_{i}=(p(s_{i}),g_{i})$ denote the element which embodies the basic deployment information of the sensor $s_{i}$, where $p(s_{i})$ is the position of the sensor $s_{i}$, and $g_{i}$ is the validity flag. Assume that $g_{i}=1$ if the sensor $s_{i}$ is deployed, otherwise $g_{i}=0$. Thus, the deployment vector $\left[e_{1},e_{2},e_{3},\cdots ,e_{k}\right]$ could represent a deployment situation of the sensors, namely a feasible solution to SDP, where k is the maximum number of the available sensors. 

Every point in the feasible region is supposed to be revisited periodically. Therefore, the feasible region should be covered completely in SDP. Fewer sensors and less overlap of cells could cause a shorter path, and hence improve the efficiency of persistent coverage task. Taking the objectives of the sensor deployment into account, the fitness function of the sensor deployment is defined as:(11)$$f_{s}(S)=\lambda_{1}\times n_{b}+\lambda_{2}\times n_{u}+\lambda_{3}\times n_{s}+\lambda_{4}\times n_{o}$$
where S indicates a feasible solution to SDP; $n_{b}$, $n_{u}$, $n_{s}$, and $n_{o}$ are the number of the sensors deployed in the obstacles, the number of the uncovered cells, the number of sensors deployed, and the number of the overlapped cells, respectively; $\lambda_{1}$, $\lambda_{2}$, $\lambda_{3}$, and $\lambda_{4}$ represent the corresponding weighted factors, which satisfy $\lambda_{1}\gg \lambda_{2}\gg \lambda_{3}>\lambda_{4}>0$. 

The fitness function is minimized based on the improved CCPSO2 algorithm to obtain the optimal deployment vector. The CCPSO2 is adopted due to its ability for solving large-scale optimization problems. Considering the properties of SDP, several heuristic operators are introduced to promote the performance of the normal CCPSO2. As depicted in Figure 6, Figure 6a shows the sensor deployment using improved CCPSO2 with several heuristic rules and Figure 6b illustrated the deployment situation using standard CCPSO2 with no heuristic rules. There are 99 sensors used in Figure 6a while a total of 103 sensors are used in Figure 6b. According to the cost curves for two cases in Figure 6c, it can be found that the introduction of heuristic rules can effectively improve the convergence speed of the CCPSO2. In addition, the heuristic operators can also improve the diversity of the particles, and hence improve the performance of the algorithm in solving SDP. Submitting (3) to (6), the pseudo-code for sensor deployment is listed in **Algorithm 1**.

**Algorithm 1.** Pseudo-code of improved cooperatively coevolving particle swarm optimization (CCPSO2) for sensor deployment.**Algorithm 1**: Improved CCPSO2 for sensor deployment.01:// **Initialization**: 26:  // Update particle by heuristic rules02:Set task parameters.27:  **if**
rand<p03:Set parameters of improved CCPSO2.28:  Remove the sensor with closer distance via **Deletion** operator.04:Randomly initialize K swarms, each with s sensors.29:  **if** the task area is covered completely05:// **Main loop**:30:   Fuse and move sensor by **Fusion** and **Movement** operators.06:**Repeat** each iteration31:  **end**07: **if**$f_{s}(\hat{y})$ has not been improved,32:  **else**08: Choose value s from a predefined set randomly, contrast K (K=n/s).33:   Add and move sensor by **Addition** and **Movement** operators09: **end**34:  **end**10: // Update optimal position35:  **end**11: **for** each swarm j
36:  // Update particle by CCPSO2 rules12: **for** each particle i37:  **else**13:  $P_{j}.y_{i}$ and $P_{j}.{\hat{y}}_{i}$ are updated as (6).38:  **if**
rand<q14: **end**39:   The ith particle is updated as (2).15: **for** each particle i40:  **end**16:  $P_{j}.{y^{\prime }}_{i}$ is updated as (5).41:  **else**17: **end**42:   The ith particle is updated as (3).18: Compute fitness value of $b(j,P_{j}.x_{i})$
 and ${y^{\prime }}_{i}$.43:  **end**19: **if**
$f_{s}(b(j,P_{j}.x_{i}))<f_{s}({\hat{y}}_{i})$44:  **end**20:  $b(j,P_{j}.x_{i})=P_{j}.{\hat{y}}_{i}$45: **end**21: **end**46: **end**22: **end**47:
**end**
23: // Update particles in each swarm48:// **Results:**24: **for** each swarm j
49:Find the optimal solution as the sensor deployment situation.25: **for** each particle i

In this paper, the sensors are deployed to cover the sub-region with the smallest coverage period first, then the sub-region with larger coverage period, and so on. As shown in Figure 2, there are three sub-regions in the task area. The task coverage period satisfies $T_{1}>T_{2}>T_{3}$. This means that sub-region 1 is more important than sub-region 2, and sub-region 2 is more important than sub-region 3. Therefore, the order of sub-regions to be covered by the sensors is [sub-region 1, sub-region 2, and sub-region 3]. After determining the coverage order, each sub-region is sequentially covered according to **Algorithm 1**. The sub-regions that have been covered by the sensors in the former phases are seen as obstacles when deploying sensors over other sub-regions in the latter phase. It should be noted that the sensor deployment of each sub-region is optimized independently. In this way, it can reduce the complexity of the algorithm and speed up the convergence process, as well as facilitate partition of closed path in subsequent work.

### 4.3. Path Planning Based on Travelling Salesman Problem

Taking positions of deployed sensors as the waypoints, TSP is solved to obtain the frame of the path for persistent coverage. The feasible region can be covered completely if a robot circumnavigates the closed path once. 

Let $r_{n}$ be the neighboring range and $p(s_{i})$ be the position of the sensor $s_{i}$. Taking waypoints as vertices, we define the proximity graph G=(V,E), where V and E denote the vertices and edges. E is defined as $E=\left\{(s_{i},s_{j})\in V\times V:\left\|p(s_{i})-p(s_{j})\right\|\leq r_{n},s_{i}\neq s_{j}\right\}$, where $\left\|p(s_{i})-p(s_{j})\right\|$ is the Euclidean distance between the vertices $s_{i}$ and $s_{j}$. A path that connects the vertices $s_{1}$ and $s_{n}$ is defined as a sequence of the distinct vertices $sq_{i}(s_{1},s_{n})=[s_{1},s_{2},\cdots ,s_{i},\cdots ,s_{n}]$, which satisfies $(s_{i},s_{i+1})\in E$, 1≤i≤n−1. The graph G is connected if there is a path between any two distinct vertices. If the feasible region is covered by the deployed sensors completely and the neighboring range $r_{n}$ is no smaller than twice the virtual sensor radius $r_{v}$, that is $r_{n}\geq 2r_{v}$, the proximity graph G is connected [44]. A path that connects the sensors $s_{i}$ and $s_{j}$ is illustrated in Figure 7, where the neighboring range $r_{n}=2r_{s}$.

Let $sr(s_{i})$ denote the sub-region in which the sensor $s_{i}$ is deployed. The length of the path $sq_{i}(s_{1},s_{n})$ is defined as:(12)$$l(sq_{i}(s_{1},s_{n}))=\sum_{i=1}^{n-1}\left(\left\|p(s_{i})-p(s_{i+1})\right\|\right)+\sum_{i=1}^{n-1}\alpha I(sr(s_{i})\neq sr(s_{i+1}))$$
where $\left\|p(s_{i})-p(s_{i+1})\right\|$ is the Euclidean distance between the vertices $s_{i}$ and $s_{i+1}$; α>0 is a weighted constant; Ι(⋅) denotes the indicator function, which satisfies:(13)$$I(sr(s_{i})\neq sr(s_{i+1}))=\left\{ \begin{array}{l} 1,sr(s_{i})\neq sr(s_{i+1})\\ 0,sr(s_{i})=sr(s_{i+1}) \end{array}\right.$$

In order to reduce the connections between different sub-regions in the frame of the path, instead of the Euclidean distance, the distance between two sensors $s_{1}$ and $s_{n}$ is defined as:(14)$$d(s_{1},s_{n})=\underset{sq_{i}(s_{1},s_{n})}{\min}\left(l(sq_{i}(s_{1},s_{n}))\right),sq_{i}(s_{1},s_{n})\in SQ(s_{1},s_{n})$$
where $SQ(s_{1},s_{n})$ is the set of the paths that connect the sensors $s_{1}$ and $s_{n}$. 

In the task of cooperative persistent coverage, the path frame needs to be divided into several segments. What is more, fewer connections among different sub-regions would be beneficial to reduce the number of segments, i.e., the number of the robots. Therefore, in order to facilitate path segmentation in subsequent work, it is necessary to reduce connections among different sub-regions as much as possible. The frames obtained based on different distance definitions are presented in Figure 8. The Euclidean distance is used for path planning in Case 1. It can be seen that the frame spanned eight times between sub-region 1 and sub-region 2. In addition, the path frame is divided into eight segments by the boundaries. Among them, there are four segments in sub-region 1, and four segments in sub-region 2. While in Case 2, the distance between two sensors is defined by (14) for path planning. It is clear that the frame only crosses twice between sub-region 1 and sub-region 2. In the same time, the path is divided into two sections, each sub-region with a section. Obviously, the frame in Case 2, effectively reduces the connections in different sub-regions, and the planning result is more suitable for the path segmentation.

The shortest path that connects any two waypoints as well as its length is obtained based on the A-star algorithm. The evaluation function is in the form of: (15)$$f_{a}(M)=g(M)+h(M)$$
where g(M) refers to the actual cost function of the path that has passed, from the start point to the current point M; h(M) indicates the heuristic function of the remaining path, from the current point M to the target point. g(M) can be formulated as:(16)$$g(M)=\tau_{1}\cdot f(M-1)+\tau_{2}\cdot D_{cur}+\tau_{3}\cdot P_{obs}$$
where f(M−1) is the cost value from the start point to the M−1th point; $D_{cur}$ denotes the distance between previous point M−1 and current point M; $\tau_{1}$, $\tau_{2}$, and $\tau_{3}$ are the weighting coefficients of each item. The heuristic function can be estimated by the following expression:(17)$$h(M)=\tau_{4}\cdot D_{res}$$
where $D_{res}$ represents the distance from current point M to the target point; $\tau_{4}$ is the weighting coefficient.

In this paper, modified GA is used to solve TSP on account of its inherent parallelism and global search capability. The fitness function can be defined by:(18)$$f_{t}(sq_{i}(s_{1},s_{n}))=\frac{1}{l(sq_{i}(s_{1},s_{n}))+D(s_{n},s_{1})}$$
where $sq_{i}(s_{1},s_{n})$ refers to the ith feasible path; $D(s_{n},s_{1})$ indicates the distance between the city $s_{n}$ and $s_{1}$, which satisfies:(19)$$D(s_{n},s_{1})=\left\|p(s_{n})-p(s_{1})\right\|+\alpha I(sr(s_{n})\neq sr(s_{1}))$$
where $\left\|p(s_{n})-p(s_{1})\right\|$ is the Euclidean distance between the city $s_{n}$ and $s_{1}$; α>0 is a weighted constant; Ι(⋅) denotes the indicator function. Then, the TSP can be formulated by the modified GA as follows.

**Algorithm 2.** Pseudo-code of modified genetic algorithm (GA) to solve the travelling salesman problem (TSP).**Algorithm 2**: Modified GA to solve TSP.01:// **Initialization**: 13: **switch**
k02:Obtain sensor locations from the result of **Algorithm 1**.14:  **case 1** No operation on tem_pop(1)03:Set parameters of modified genetic algorithm.15:  **case 2** Perform **Mutation operation** on tem_pop(2)
04:Randomly initialize 4×K populations.16:  **case 3** Perform **Evolutionary reversal**
**operation** on tem_pop(3)
05:// **Main loop**:17:  **case 4** Perform **Slide operation** on tem_pop(4)
06:**Repeat** each generation18: **end**07: Calculate individual fitness values for each population.19: **end**08: **Selection operation**: Search the individual with the highest fitness value.20: Use tem_pop to generate 4×K new populations. 09: Update optimal individual opt_pop.21:
**end**
10: // Generate new populations22:
**// Results:**
11: **for**
k=1:423:Get the best path found by the algorithm.12:  tem_pop(k)=opt_pop



### 4.4. Partition of Closed Path Considering Coverage Period

Given the frame obtained by solving the travelling salesman problem, the path for complete coverage can be obtained by circumnavigating the frame, as depicted in Figure 9a. The blue line is the frame of the closed path, which connects all the waypoints in the mission area. The red closed route is generated on the basis of the blue frame and robot’s sensing range. The robot carrying detection sensor with sensing radius $r_{s}$ circumnavigates the red route to achieve full coverage of the current region.

In order to satisfy the coverage period constraints, multiple robots may be required. Taking the coverage periods of the sub-regions into account, the frame is partitioned into several segments with the aim of minimizing the number of the segments. On the basis of each partitioned segment, the closed route is generated for each robot. Afterwards, each robot is assigned one closed route to circumnavigate in order to achieve persistent coverage of the feasible region. Assume that $\gamma_{i}$ refers to the normalized length between the sensor $s_{i}$ and sensor $s_{i+1}$. It can be determined by the following expression:(20)$$\gamma_{i}=\frac{d(s_{i},s_{i+1})}{D_{sum}},\gamma_{i}\in [0,1]$$
where $d(s_{i},s_{i+1})$ is the actual length of the ith segment; $D_{sum}$ represents the length of the frame of the closed path. Total path length $D_{sum}$ can be given by:(21)$$D_{sum}=\sum_{i=1}^{n}d(s_{i},s_{i+1})$$
where n is the number of sensors deployed; the sensor $s_{n+1}$ refers to the sensor $s_{1}$, which satisfies $d(s_{n},s_{n+1})=d(s_{n},s_{1})$. 

Let the frame of the closed path be denoted as a closed curve. The curve equation can be expressed as r(ϑ):[0,1]→ℝ2, r(0)=r(1), where ϑ is the normalized length of the curve. r(0) denotes the initial point of the curve, which is the location of sensor s1. Simultaneously, r(0) is also the initial point of curve segmentation. Similarly, r(1) is the end point of the curve. Since the curve is closed, there is r(0)=r(1). It should be noted that the curve equation r(ϑ) is a function of the normalized length ϑ. 

Suppose that the curve is partitioned into m segments by $\left[\theta_{1},\theta_{2},\cdots ,\theta_{m}\right]$, where $\theta_{1}<\cdots <\theta_{m}$. Param $\theta_{i}$ represents the normalized length between the initial point r(0) and the termination point $r(\theta_{i})$ of the ith segment, which is defined as:(22)$$\theta_{i}=\sum_{k=1}^{K}\gamma_{k}\;,\;1\leq k\leq K,\;1<K\leq n$$
where $\gamma_{k}$ is the normalized length between the sensor $s_{k}$ and sensor $s_{k+1}$; K refers to the serial number of the last sensor contained in the ith segment; n is the number of sensors. 

Let ri(ϑ) be the ith segment, which can be described as ri(ϑ):(θi−1,θi]→ℝ2, 1≤i≤m, where θi is the normalized length between the r(0) and the termination point r(θi) of the current segment; r(θm) refers to the end point r(1) of the curve; r(θ0) represents the initial point r(0) of the curve. It should be noted that the following formula is established r(θm)=r(1)=r(0)=r(θ0). Additionally, r(θi) and ri(θi) are the same point, which satisfies ri(θi)=r(θi)=p(sK), where p(sK) denotes the position of the last sensor in the ith segment. 

Therefore, the problem of frame partition is turned into finding the optimal termination point $r(\theta_{i})$ for each segment according to the coverage period constrains. The point $r(\theta_{i})$ also represents the best location $p(s_{K})$ of the last sensor $s_{K}$ contained in each segment. Figure 9b illustrates an example of path partition. The closed path is divided into four segments: $r_{1}(\vartheta )$, $r_{2}(\vartheta )$, $r_{3}(\vartheta )$, and $r_{4}(\vartheta )$. There are eighteen sensors in the closed path. Among them, segment 1 contains four sensors from $s_{2}$ to $s_{5}$; segment 2 involves four sensors from $s_{6}$ to $s_{9}$; segment 3 contains five sensors from $s_{10}$ to $s_{14}$; segment 4 includes five sensors from $s_{15}$ to $s_{18}$. 

Let $T^{c}(s_{j})$ be the coverage period of the sub-region where the sensor $s_{j}$ is deployed. Let $S_{i}$ be the set of the sensors which are deployed on the segment $r_{i}(\vartheta )$, where ϑ satisfies $\vartheta \in (\theta_{i},\theta_{i+1}]$. The coverage period of the ith segment $r_{i}(\vartheta )$ can be written in the equivalent form as follows:(23)$$T_{i}^{c}=\underset{s_{j}\in S_{i}}{\min}(T^{c}(s_{j}))$$

Assume $R_{c}(r_{i}(\vartheta ))$ is the closed curve generated by the segment $r_{i}(\vartheta )$ based on computational geometry. $R_{c}(r_{i}(\vartheta ))$ also represents the closed route assigned to each robot. As shown in Figure 10, red line refers to the closed route generated by the blue segment $r_{i}(\vartheta )$. The width of the closed route is twice the sensing radius of each robot, that is $2r_{s}$. When the obstacle blocks the closed route, the method will re-adjust the closed route according to the threat range of the obstacle.

The purpose of the frame partition is to minimize the number of robots performing tasks, while meeting coverage period constraints of different sub-regions. Consequently, the objective of the frame partition problem can be described as:(24)minm
subject to:(25)$$T_{i}^{a}<T_{i}^{c},\;\forall i\in \{1,2,\cdots ,m\}$$
where $T_{i}^{a}$ is the time required for a robot to circumnavigate the closed route $R_{c}(r_{i}(\vartheta ))$ once. $T_{i}^{a}$ can be obtained by the length of the closed path and robot’s velocity. It is given in the following expression as:(26)$$T_{i}^{a}=\frac{L(R_{c}(r_{i}(\vartheta )))}{V},\vartheta \in (\theta_{i-1},\theta_{i}]$$
where $L(R_{c}(r_{i}(\vartheta )))$ denotes the length of the closed route $R_{c}(r_{i}(\vartheta ))$; V is the velocity of robot. It is assumed that the robots move at a constant speed for simplicity. 

The frame partition problem is solved based on the PSO due to its flexibility and global optimization capability [45]. The update of velocity and position are determined by:(27)$$\left\{ \begin{array}{l} v_{i}(t+1)=\omega (t)v_{i}(t)+c_{1}r_{1}(t)(y_{i}(t)-x_{i}(t))+c_{2}r_{2}(t)(\hat{y}(t)-x_{i}(t))\\ x_{i}(t+1)=x_{i}(t)+v_{i}(t+1) \end{array}\right.$$
where $v_{i}(t)$ and $x_{i}(t)$ represent the velocity and position of the ith particle respectively; $y_{i}(t)$ is the local best position for the ith particle; $y_{i}(t)$ denotes the global optimal position of particles in the swarm; ω(t) is the inertial weight; $c_{1}$ and $c_{2}$ are acceleration constants; $r_{1}(t)$ and $r_{2}(t)$ refer to the random numbers, which satisfy $r_{1}(t)\in [0,1]$, $r_{2}(t)\in [0,1]$. 

In the path partition algorithm, actual coverage time $T_{i}^{a}$, coverage period constraint $T_{i}^{c}$ and the number of segments $N_{seg}$ are the main considerations for the frame partition of closed path. Thus, the fitness function can be defined as:(28)$$f_{p}(\chi )=\xi_{1}\times T_{dif}+\xi_{2}\times N_{seg}+\xi_{3}\times T_{err}$$
where χ is a feasible solution to the frame partition; $T_{dif}$ indicates the time difference between the actual coverage time $T_{i}^{a}$ and coverage period constraint $T_{i}^{c}$, which satisfies $T_{con}=T_{i}^{a}-T_{i}^{c}+T_{m}$; $T_{err}$ refers to the time offset between $T_{i}^{a}$ and $T_{i}^{c}$, which satisfies $T_{err}=T_{i}^{c}-T_{i}^{a}-T_{m}$; $\xi_{1}$, $\xi_{2}$, and $\xi_{3}$ are the weighting coefficients of each item. Constant $T_{m}$ represents the time margin, which can adjust the time offset to meet the coverage period requirements. Feasible solution χ can be expressed as $\chi =\{r(\theta_{1}),r(\theta_{2}),\cdots ,r(\theta_{m})\}$, where $r(\theta_{i})$ is the termination point of the ith segment; m is the number of segments. In the path partition algorithm, the PSO optimizes a set of locations, which are the positions of the last sensor contained in each segment. In other words, the algorithm uses the feasible solution χ to characterize the position $x_{i}(t)$ in the PSO. In summary, the algorithm can be described as follows.

**Algorithm 3.** Pseudo-code of particle swarm optimization (PSO) for path partition.**Algorithm 3**: Modified GA to solve TSP.01:// **Initialization**: 15: **if**
$f_{p}(y_{i}(t))<f_{p}(\hat{y}(t))$02:Obtain the closed path from the result of **Algorithm 2**.16:  $\hat{y}(t)=y_{i}(t)$03:Set parameters of PSO.17: **end**04:Randomly initialize K swarms, each with M particles.18:
**end**
05:// **Main loop**:19:Obtain the optimal solution, as well as determine the number of segments and calculate the starting and ending position of each segment.06:**Repeat** each iteration07: Compute inertial weight ω(t).08: **for** each particle i
09:  The ith particle is updated as (24).20:Generate the closed route for each robot on the basis of portioned segment and robot’s sensing range using geometry method.10:  Calculate fitness value of each particle $f_{p}(x_{i}(t))$.11:  **if**
$f_{p}(x_{i}(t))<f_{p}(y_{i}(t))$21:// **Results:**12:  $y_{i}(t)=x_{i}(t)$22:Get the closed route for each robot’s persistent coverage.13:  **end**14: **end**



### 4.5. The Framework of Combinatorial Method

The combinatorial method proposed in this paper, is mainly composed of the above three algorithms, namely **Algorithm 1**, **Algorithm 2**, and **Algorithm 3**. As depicted in Figure 11, the framework of combinatorial approach is proposed for multi-robot persistent coverage.

The main principle is summarized as follows. Initially, set task parameters for persistent coverage, including coverage domains, sub-region size, obstacles’ location, and so on. Then, **Algorithm 1** is used to sequentially cover the sub-regions using virtual sensors, in order of mission importance. Extract the locations of each sensor, when the entire task area is completely covered and deployment cost is minimal. Furthermore, taking the positions of the deployed sensors as the waypoints, **Algorithm 2** solves the TSP to obtain the frame of the closed path which connects all the sensors. Additionally, in consideration of coverage period constraints and the optimum quantity of robots, the frame is partitioned into several segments using **Algorithm 3**. Finally, generate the closed route for each robot on the basis of portioned segment and robot’s sensing range using geometry method. Therefore, each robot can circumnavigate one closed route to cover mission domains completely and persistently. Figure 12 further illustrates the solution process of the combinatorial method for multi-robot persistent coverage.

## 5. Numerical Results

In view of the facts that previous works neglected regarding the coverage period constraints in different sub-regions and the number of robots used for persistent coverage task, this paper proposes the above combined method for cooperative multi-robot persistent coverage. In this section, the numerical results are presented to demonstrate the proposed approach. Simulation is implemented in Matlab. All algorithms are written by ourselves and we do not use any toolbox.

### 5.1. Multi-robot Persistent Coverage

The mission area is assumed to be a square with side length 120 m. There are two rectangular obstacles and two sub-regions with smaller coverage periods in the region of interest, as depicted in Figure 2. The coverage period constraints for each sub-region are shown in Table 1. Suppose that the velocity of each robot is a constant value of 20 m/s. Table 2, Table 3, and Table 4 list the parameters in the combinatorial scheme for cooperative persistent coverage. 

Figure 13 illustrates the results of the path planning for multi-robot persistent coverage, in allusion to the task area situation shown in Figure 2. It can be found that the proposed method can achieve persistent coverage for the mission area using a minimum number of robots, while avoiding obstacles in complex environments. Table 5 lists the coverage periods of each partition and actual time required to circumnavigate the closed route in Figure 13. 

As shown in Figure 13a, the mission area is completely covered by 96 virtual sensors, whose sensing radius is 10. Among them, there are 10 sensors in the sub-region 1, 15 sensors in the sub-region 2, and 71 sensors in the sub-region 3. Figure 13b shows the evolution of the fitness function of the sensor deployment. The optimal cost of its initial swarm is 5743. At last, the fitness value stabilized at 3600 after the 139 iterations. The path planning result for 96 waypoints is illustrated in Figure 13c. If a robot circumnavigates the closed path once, the task area can be covered completely. The history of closed path length is presented in Figure 13d. The closed path length tends to be stable after the 1722 generation, reaching the minimum of 61,025.99 by the 7113 generation. The value 61,025.99 is the minimum length calculated by (12), and the actual distance of the closed path is 1027.40. As depicted in Figure 13e, there are five closed routes generated for robots on the basis of portioned segment and robot’s sensing range. Further, each closed route is assigned to one robot. It can be found that each robot can track its respective closed route to cover the task area completely and persistently. Figure 13f illustrates the changing process of cost function defined by (28). The path partition cost reaches a minimum of 5709.23 after the update of 168th iteration.

### 5.2. Simulation Results Under Different Coverage Periods

To justify the effectiveness of the proposed combinatorial scheme, this part presents some simulation results with different coverage period constraints. As depicted in Figure 14, there are six simulation results of the path planning for multi-robot persistent coverage, under different coverage period constraints. The simulation parameters used in Figure 14 are the same as in Section 5.1. Table 6 lists the coverage period $T_{i}^{c}$ and actual time $T_{i}^{a}$ required for each robot to circumnavigate its own closed route in allusion to different coverage period constraints. It can be found from the simulation results that the actual coverage time used of each robot satisfies the requirements of the coverage period constraints. In addition, the combined method can adaptively adjust the number of closed routes according to the coverage period constraints. Therefore, the approach can achieve optimization of the number of robots used.

There are five closed routes in Figure 14a. It is found from the Case 1 in Table 6 that each generated closed route meets the requirements of the coverage period constraints, namely $T_{i}^{a}<T_{i}^{c}$. As shown in Figure 13, it takes about 37.50 s for two robots to circumnavigate the closed routes in sub-region 3. Similarly, it takes 18.81 s for one robot to circumnavigate the closed routes in sub-region 2. If coverage period constraints are changed to the Case 2, the simulation result is shown in Figure 14b. In such case, the approach may generate one closed route for sub-region 2 according to the above situation. However, if this is done, the combined method will generate four closed routes for sub-region 3, because coverage period constraints of sub-region 3 in Case 2 do not satisfy the actual coverage time of sub-region 3 in Figure 13. As a result, the total routes will become six. In contrast, the technique increases the number of routes in sub-region 2 to satisfy the coverage period constraints of the task area. In this way, the total routes will become five as in Case 2. Obviously, the number of routes in Figure 14b is one less. It can be found that the combinatorial approach is able to allocate an optimal number of robots to perform persistent coverage. Figure 14c–e show the planning results for persistent coverage under corresponding coverage period constraints. It can be easily found from Figure 14e and Table 6 that the actual coverage time from the 8th route to 11th route in Case 6 is slightly higher than the coverage period constraints. The reason is that the closed routes of other regions cannot compensate for the length of routes in sub-region 3, under the premise of satisfying their own coverage period constraints. In addition, if it is simply to meet the coverage period requirements, the cost of adding a closed route to sub-region 3 is much greater than the cost of Case 6. Therefore, the method sacrifices the coverage periods of some routes for the minimum coverage cost. In other words, the combined approach can tolerate the loss of coverage accuracy to obtain greater coverage gain. 

### 5.3. Cooperative Coverage in More Complex Mission Environments

To demonstrate the versatility of the combined approach, the task area is reset using special or irregular graphics. As depicted in Figure 15, the mission area is a pentagon. There are three sub-regions and two obstacles in the task area. Sub-region 1 is a hexagonal area and sub-region 2 is a circular area. The rest of the feasible region is denoted as the sub-region 3. The coverage periods of each sub-region satisfy $T_{1}>T_{2}>T_{3}$. 

Assume that the velocity of each robot is a constant value of 10 m/s. Table 7 lists the parameters for Algorithm 1 in Figure 15. The parameters used for Algorithm 2 and 3 are the same as in Section 5.1. In allusion to the task situation shown in Figure 15, path planning results for cooperative persistent coverage are presented in Figure 16. Table 8 lists the period requirement and actual time required to circumnavigate the closed route in Figure 16. 

As shown in Figure 16a, the feasible region is completely covered by 83 virtual sensors. Among them, there are 7 sensors in sub-region 1, 5 sensors in sub-region 2, and 71 sensors in sub-region 3. Figure 16b describes the evolution of the fitness function of the sensor deployment. The optimal cost of its original swarm is 7613. At last, the fitness value stabilized at 5321. The path planning result for 83 waypoints is shown in Figure 16c. The history of closed path length is presented in Figure 16d. The closed path length tends to be stable after the 5156 generation, reaching the minimum of 61,025.99. As depicted in Figure 16e, there are seven closed routes generated for robots on the basis of portioned segment and robot’s sensing range. Further, each closed route is allocated to one robot. Each robot can track its respective closed route to cover the task area completely and persistently. Figure 16f depicts the changing process of cost function defined by (28). The path partition cost reaches a minimum of 9343.48 after the update of the 102nd iteration. It can be found that the combinatorial method can be adapted to multi-robot persistent coverage in more complex mission environments, while guaranteeing both the obstacle avoidance and coverage period constraints.

## 6. Discussion

The main idea of the paper is to develop a new path planning strategy for multi-robot persistent coverage. In the meanwhile, the proposed scheme considers the coverage period constraints of different sub-regions in the mission domains, and can achieve the collaborative persistent coverage in more complex mission environments using an optimal number of the robots, while guaranteeing both obstacle avoidance and coverage period constraints. 

On the basis of the theoretical analysis and numerical simulation, the solution strategy of proposed method can be divided into three steps. The first step is to cover the task area completely using the virtual sensors with minimum cost. According to the coverage periods of sub-regions from small to large, sensor deployment in the feasible region can be performed in several stages. The sub-region with the smallest coverage period is primarily deployed, and then the sub-region with the larger period is deployed until all the sub-regions are completely covered. After determining the coverage order, the sub-regions are sequentially covered by using **Algorithm 1**. The sensor deployment of each sub-region is optimized independently to reduce the complexity of the algorithm and speed up the convergence process. When the feasible area is completely covered, the second step is to obtain the frame by using **Algorithm 2**. Taking the position of the deployed sensor as the waypoint, a closed path connecting all waypoints can be acquired by solving the TSP. Moreover, the shortest path that connects any two waypoints is obtained based on the A-star algorithm. The proposed approach introduces the idea of sensor deployment to divides path planning issues into SDP and TSP in order to effectively cope with the planning puzzle caused by environmental obstacles. Simulation results show that the method can effectively improve the adaptability and robustness of the algorithm to the more complex environment. Given the closed frame, the third step is to generate the closed route for each robot by using **Algorithm 3**. Before generating the closed routes, it is first necessary to divide the path frame into several segments on the basis of the coverage period constraints. Actually, the purpose of the frame partition is to divide the frame into the least number of segments under the premise of satisfying the coverage period constraints. Afterwards, the computational geometry technique is utilized to generate the closed route for each robot on the basis of each partitioned segment. Finally, each closed route is allocated to one robot to circumnavigate periodically in order to achieve persistent coverage of the feasible region. Numerical results indicate that the combinatorial approach is able to allocate an optimal number of robots to perform persistent coverage and can adaptively adjust the number of robots used according to the coverage period constraints. In conclusion, the persistent coverage achieved in this paper has the following characteristics: (1) cooperative persistent coverage for multiple robots is achieved; (2) the planning puzzle caused by environmental obstacles is solved; (3) the coverage period constraints of different sub-regions are considered; (4) the number of robots used can be adaptively adjust according to the coverage period constraints; (5) the optimal number of robots is utilized to perform the persistent coverage task.

Similar works for multi-robot cooperative coverage can be found in [13,46]. Karapetyan et al. [13] present two approximation heuristics based on boustrophedon cellular decomposition for solving the multi-robot complete coverage. The first algorithm is a direct extension of the work of Xu et al. [10] for multi-robot systems. The solution process can be divided into three steps. The approach first partitions the task area into nonoverlapping cells. Then it solves the Chinese postman problem to find a single optimal route that covers these cells. Finally, the algorithm splits the resulting route between multiple robots using the k-postman approximation algorithm proposed by [47]. Different from the first method, the second scheme first divides the task area into approximately equal partitions between robots by using greedy approach, then it utilizes the coverage algorithm proposed by [10] to plan the coverage route for each sub-region. The algorithms proposed by Karapetyan et al. can commendably solve the problem of area decomposition and route planning for complex environments with obstacles, and provide a new solution for collaborative coverage problems in complex task areas. Compared with the combinatorial approach proposed in this paper, the works of Karapetyan et al. mainly have the following differences: (1) the methods proposed by [13] achieve the complete coverage of the given area, not a periodic persistent coverage; (2) the number of robots performing cooperative coverage is artificially determined and not adaptively adjusted according to task requirements; (3) the sub-regions with different task importance in the mission area are not considered, similarly the coverage period constraints of different sub-regions are neglected. Palacios-Gasόs et al. [46] develop an online algorithm for multi-robot persistent coverage in which each robot locally finds the optima paths and coverage actions to maintain the desired coverage level over the whole area. Firstly, the method divides the task area into several particular regions by using Voronoi diagrams, one for each robot, to avoid long shifts and conflicts with the other robots. Then, each robot creates a list of potential goals that includes the points of its region in which the coverage level can be improved the most. Next, the algorithm calculates the candidate paths to all potential goals from the list using the fast-marching method. Finally, the optimal path is selected to calculate the optimal coverage action and control the movement of the robot. The distributed algorithm proposed by [46] can actively select the coverage goals in a continuous environment and plan the optimal paths to such goals. Furthermore, the dynamic window approach for navigation is introduced to efficiently improve the algorithm competitive in terms of flexibility and robustness in changing environments. The work of Palacios-Gasόs et al. can implement multi-robot cooperative persistent coverage effectively and reach the requirement of the desired coverage level quickly. Similarly, the method proposed by [46] neglects the coverage period constraints of the sub-regions with different task importance in the mission area. Moreover, the number of robots used is also preset and not adaptively adjusted according to task requirements. However, Palacios-Gasόs et al. present a novel path planning for solving multi-robot persistent coverage in complex task environments. 

The proposed method in this paper is an offline method for multi-robot persistent coverage and lacks experiment results. Palacios-Gasόs et al. present a new online algorithm for solving multi-robot persistent coverage in complex task areas. The future work will focus on the online path planning for cooperative persistent coverage with reference to the work of Palacios-Gasόs et al. and give the actual experiments for proving the effectiveness of the proposed approach. In addition, future research will be also centered on the planning of dynamic sub-regions, namely, the sub-region and coverage period are not fixed by the prior information. Furthermore, we will promote the proposed algorithm by using the more advanced machine learning algorithms [48,49] for multi-robot persistent coverage. 

## 7. Conclusions

This paper presents a new combinatorial method for multi-robot persistent coverage in complex mission environments using an optimal number of robots.

(1) The proposed method achieves path planning for cooperative persistent coverage, in complex task areas. The path planning problem is decomposed into the sensor deployment problem and the travelling salesman problem. The planning technique, based on sensor deployment, effectively solves the obstacle avoidance in a complex environment. 

(2) Sub-regions with different task importance in the mission area are considered in this paper. Moreover, the combined method can adaptively adjust the number of closed routes according to the coverage period constraints of different sub-regions, while also optimizing the number of robots performing the task. 

According to the aforementioned description of the combinatorial approach, the planning results of the proposed method are highly effective for solving the cooperative persistent coverage. From the global perspective, the design approach takes the coverage period as the main basis and generates the closed routes for each robot in terms of the mission environment. Furthermore, each robot can circumnavigate one closed route to cover task domains completely and persistently.

## Figures and Tables

**Figure 1 sensors-19-01994-f001:**
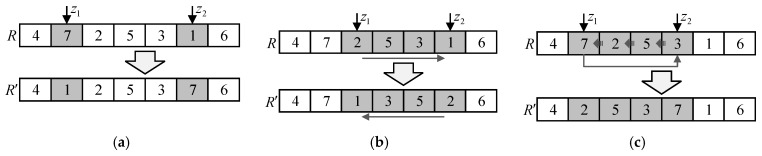
Schematic diagrams of three operations. (**a**) Mutation operation using randomly exchanging two genes of one chromosome; (**b**) Evolutionary reversal operation through re-sorting genes between two random gene positions; (**c**) Slide operation via rotating one gene position left between two random gene positions.

**Figure 2 sensors-19-01994-f002:**
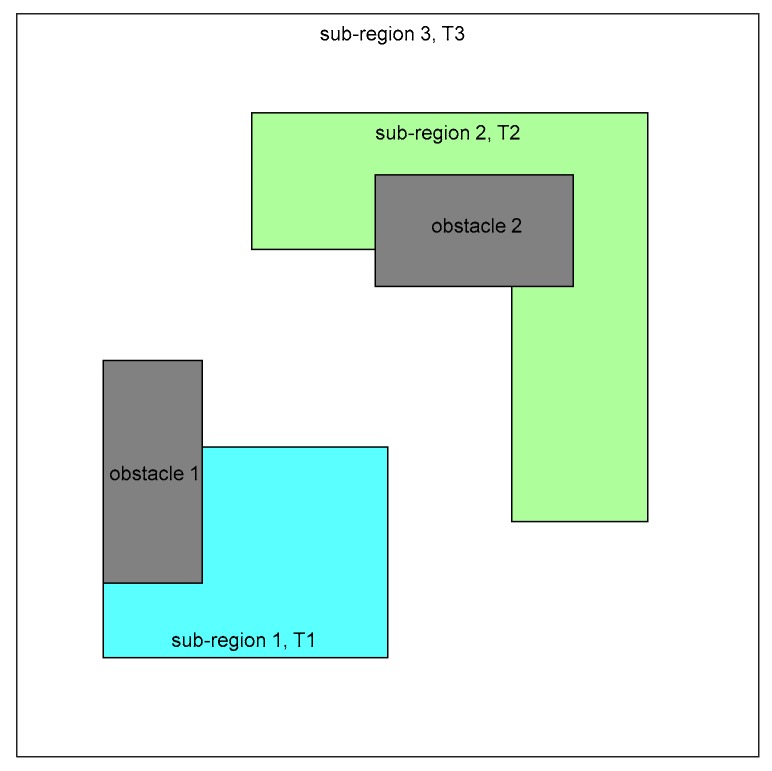
Sub-regions and obstacles in the task area. The cyan area is sub-region 1. The green area is sub-region 2. The white area is sub-region 3. Gray areas refer to obstacles.

**Figure 3 sensors-19-01994-f003:**
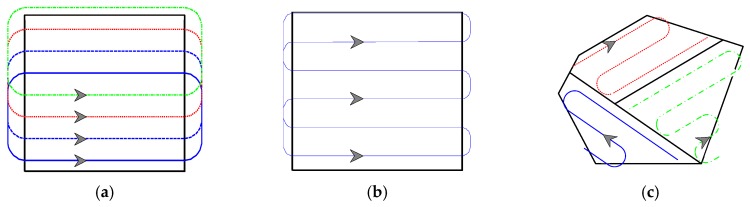
Three schematic diagrams of persistent coverage based on different methods. (**a**) An example of the parallel scanning coverage; (**b**) An example of the sequential scanning coverage; (**c**) An example of the area-decomposition based coverage.

**Figure 4 sensors-19-01994-f004:**
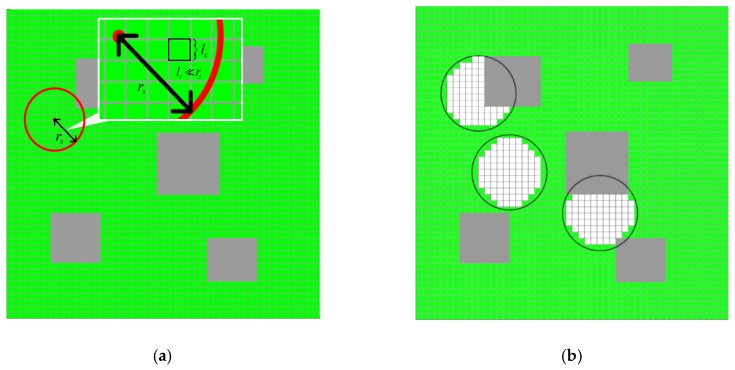
(**a**) Discretization of coverage area. The task region is divided into rectangular cells of 50 × 50. Grey rectangles refer to the obstacles. (**b**) The coverage of the sensors in the presence of obstacles. White cells indicate that they have been covered by the virtual sensors. Green cells represent that they are uncovered by any virtual sensor.

**Figure 5 sensors-19-01994-f005:**
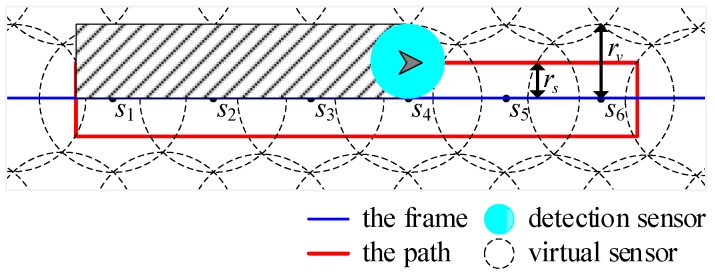
An example of sensor deployment. The radius of each robot is *r_s_* and the radius of the virtual sensor is *r_v_*. The shaded area represents the field that robots have scanned.

**Figure 6 sensors-19-01994-f006:**
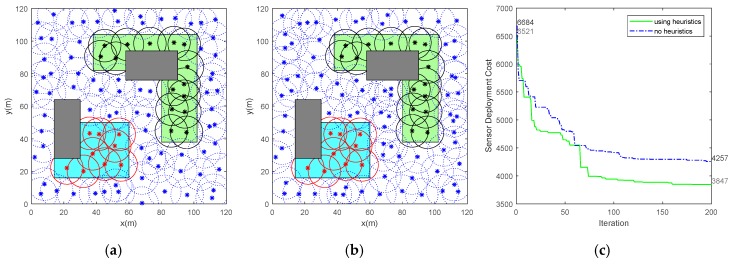
(**a**) Sensor deployment using improved cooperatively coevolving particle swarm optimization (CCPSO2) with heuristic rules. (**b**) Sensor deployment using normal CCPSO2 with no heuristic rules. (**c**) Sensor deployment cost for two situations.

**Figure 7 sensors-19-01994-f007:**
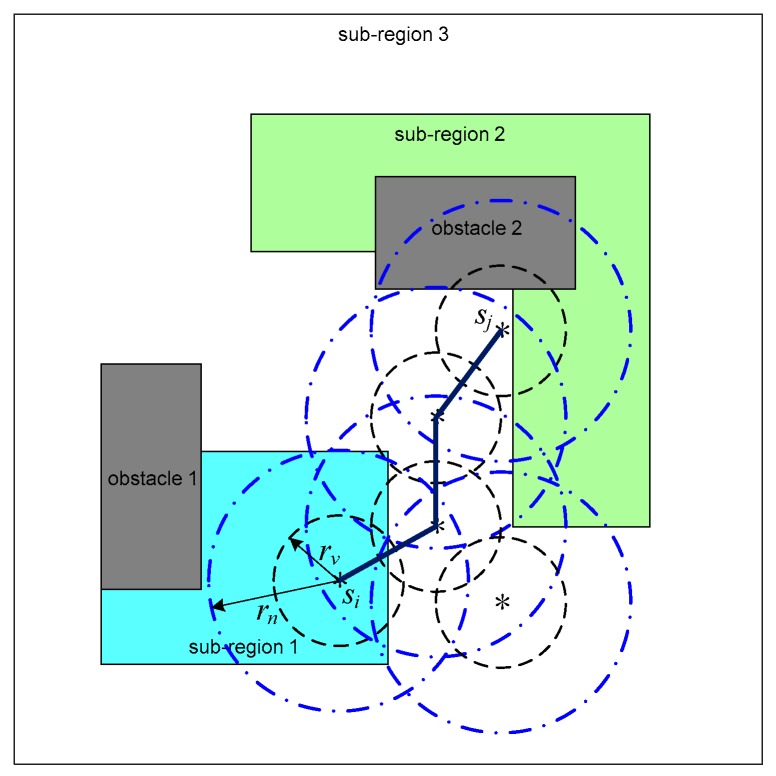
A path that connects the sensors *s_i_* and *s_j_*. Black circle is the sensing range of virtual sensor. Black asterisk refers to the position of the sensor. Blue circle denotes the neighboring radius of the virtual sensor.

**Figure 8 sensors-19-01994-f008:**
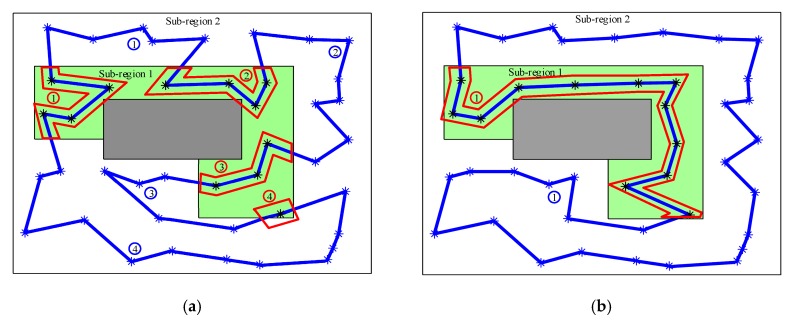
Frame generated based on different distance definitions. Grey rectangles refer to obstacles. Green area indicates sub-region 1. White area is sub-region 2. (**a**) Using Euclidean distance. (**b**) Using distance defined by (14).

**Figure 9 sensors-19-01994-f009:**
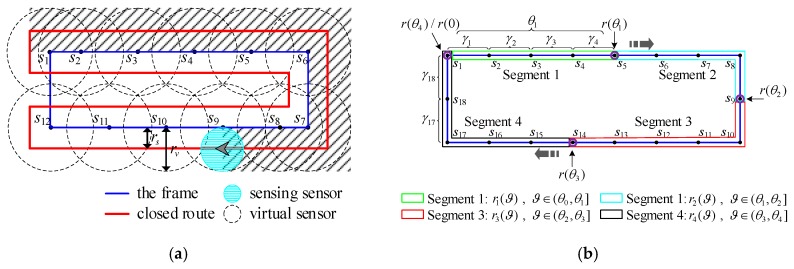
(**a**) The schematic diagram of the closed route for complete coverage generated by the frame. The radius of each robot is *r_s_* and the radius of virtual sensor is *r_v_*, which satisfies *r_v_* = 2*r_s_*. The shaded area is the field that robots have scanned. (**b**) An example of path partition. Segment 1 is inside the green box. Segment 2 is inside the cyan box. Segment 3 is inside the red box. Segment 4 is inside the black box. Termination point of each segment is inside the purple circle.

**Figure 10 sensors-19-01994-f010:**
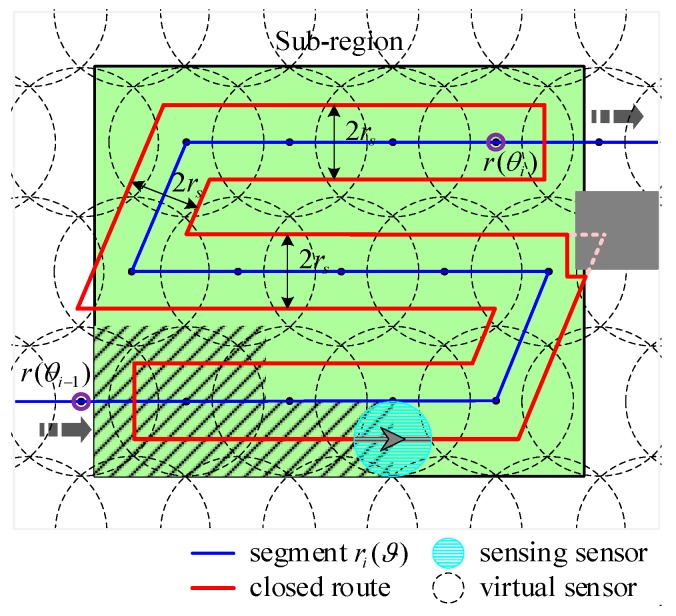
An example of the closed route generated by the segment. Green area indicates sub-region. Grey rectangle is obstacle. The shaded area is the field that robots have scanned. The termination point of each segment is inside the purple circle.

**Figure 11 sensors-19-01994-f011:**
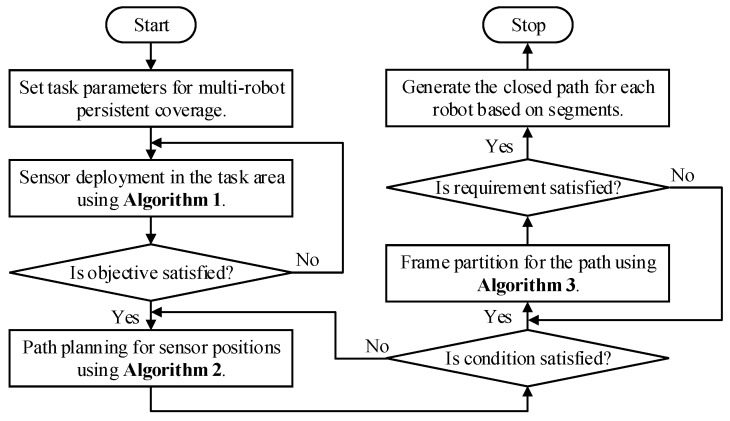
The flow chart of the combinatorial method proposed for cooperative persistent coverage.

**Figure 12 sensors-19-01994-f012:**
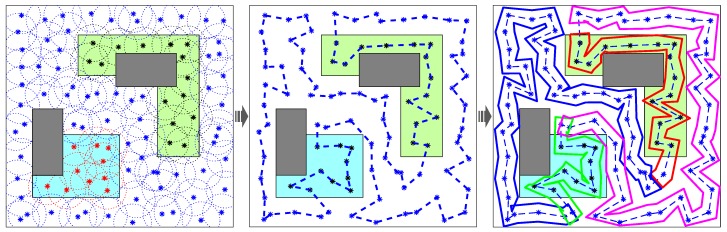
The solution process of the combinatorial method for multi-robot persistent coverage. The cyan area is sub-region 1. The green area is sub-region 2. The white area is sub-region 3. Gray areas refer to obstacles. The asterisk denotes the position of the sensor. The blue dotted line is the frame. Solid curve represents the closed route for each robot.

**Figure 13 sensors-19-01994-f013:**
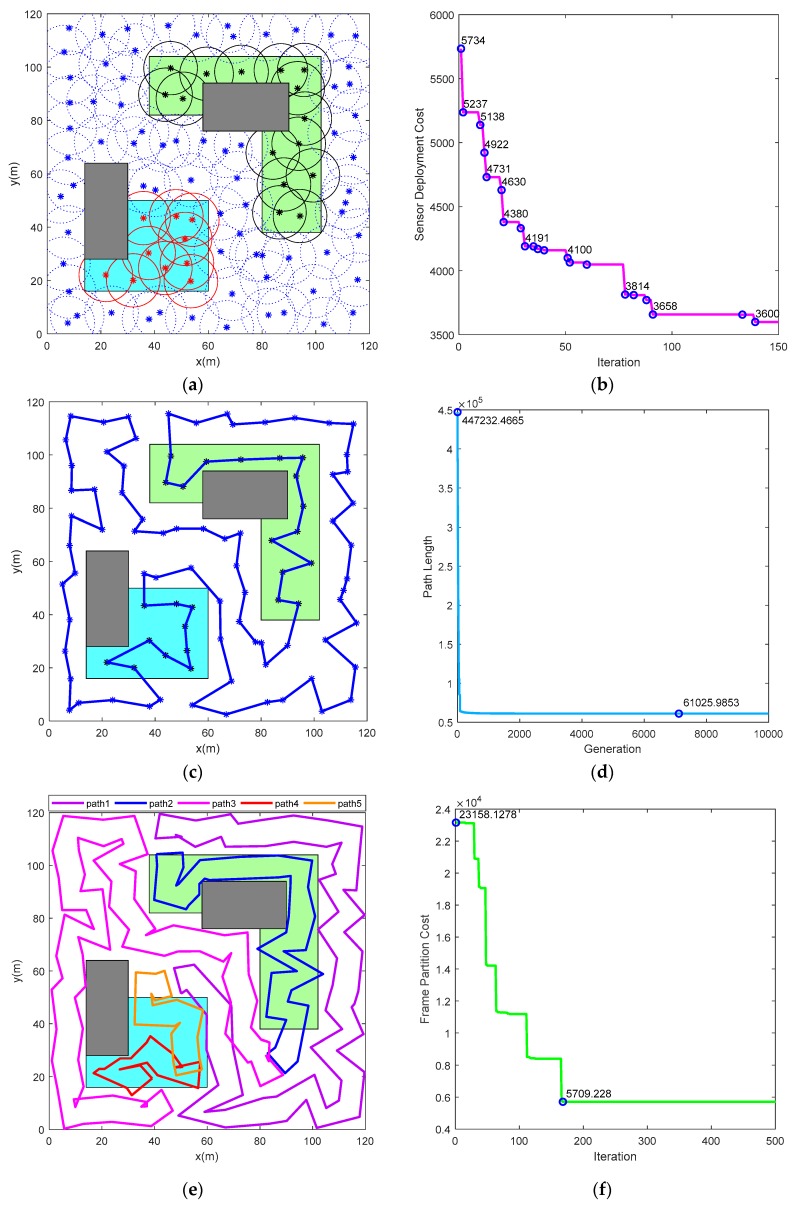
Simulation results under parameters in Table 2, Table 3 and Table 4. (**a**) Sensor deployment for complete coverage; (**b**) Sensor deployment cost; (**c**) Path planning result; (**d**) The changing process of path length; (**e**) The closed route for each robot; (**f**) Frame partition cost.

**Figure 14 sensors-19-01994-f014:**
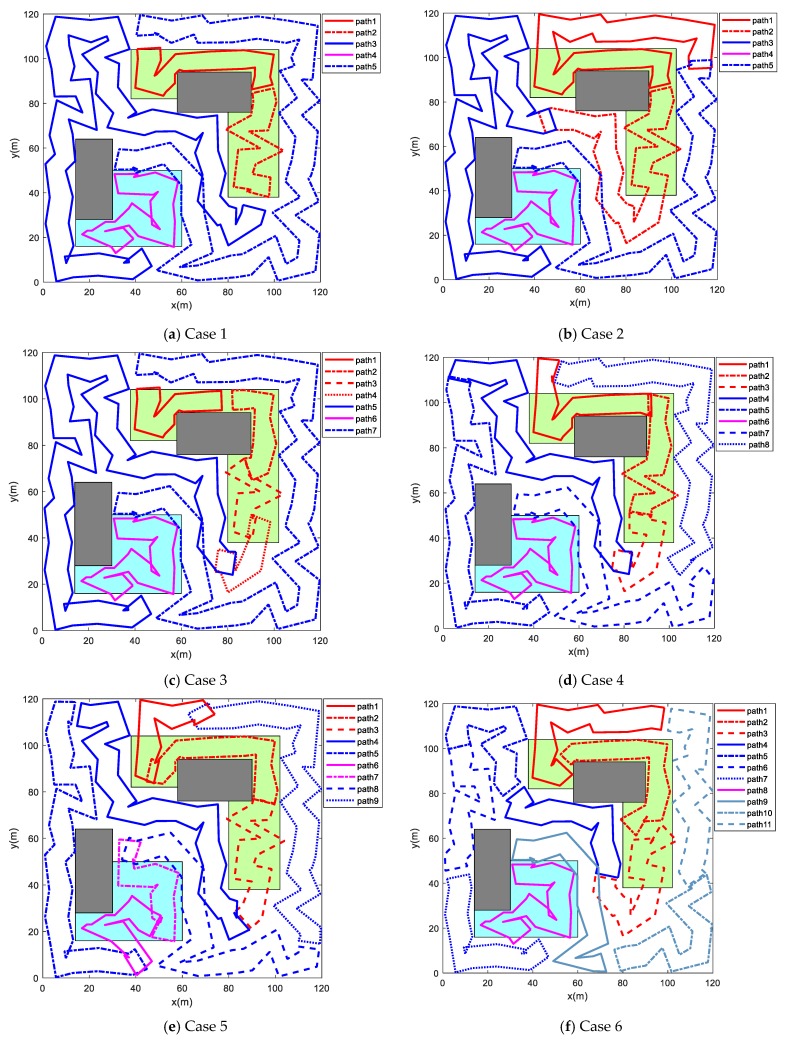
Simulation results under different coverage period constraints.

**Figure 15 sensors-19-01994-f015:**
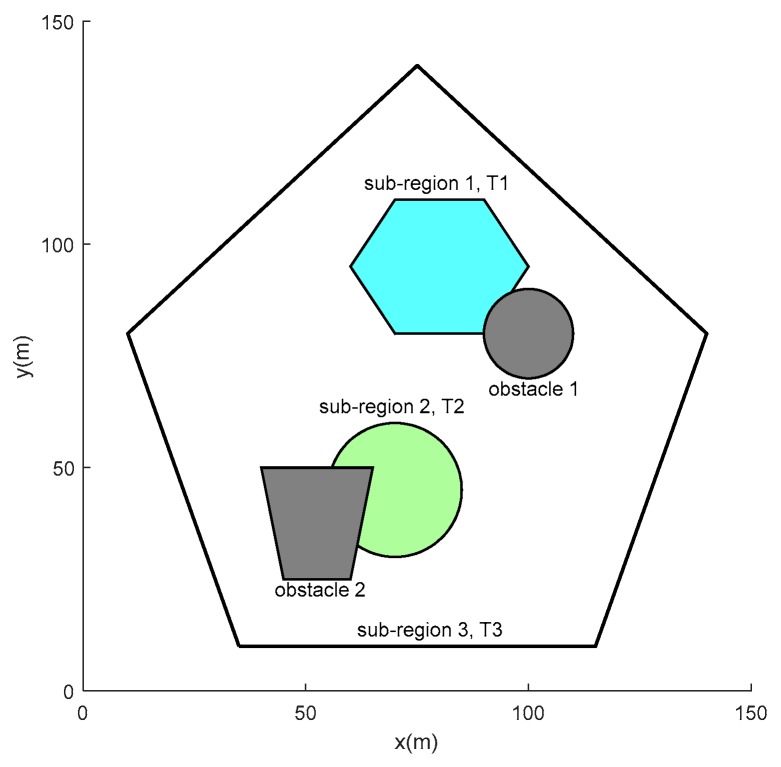
The distribution of the task environment.

**Figure 16 sensors-19-01994-f016:**
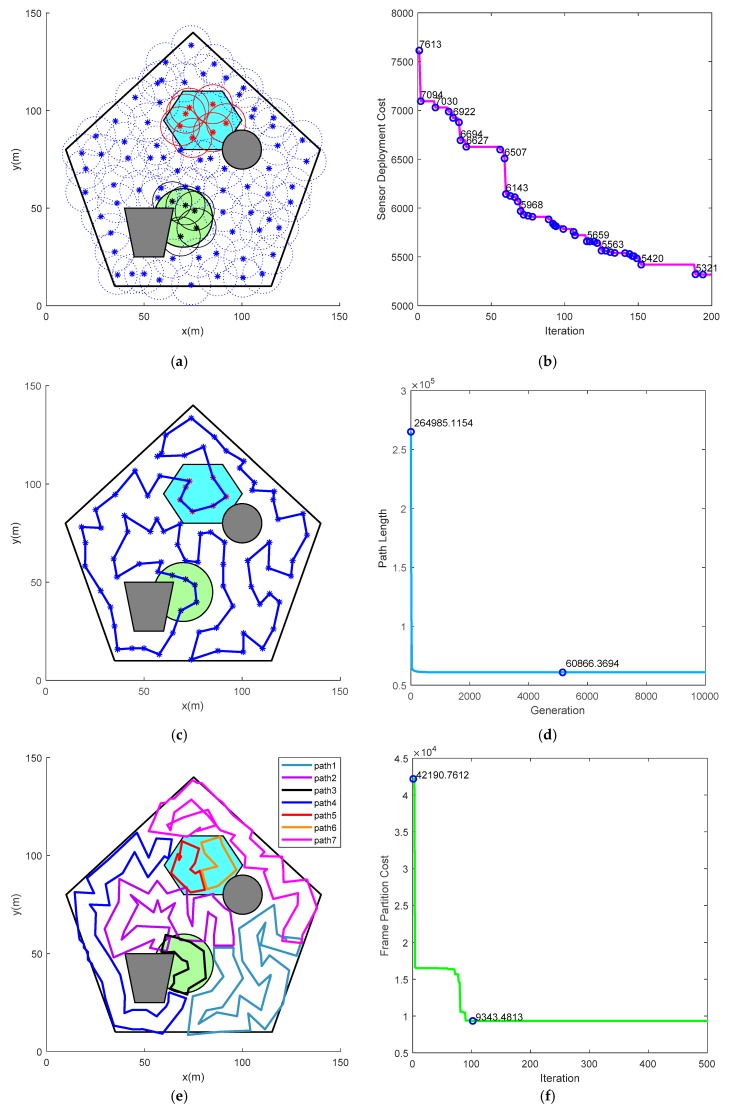
Simulation results under parameters in Table 7 and Table 8. (**a**) Sensor deployment for complete coverage; (**b**) Sensor deployment cost; (**c**) Path planning result; (**d**) The changing process of path length; (**e**) The closed route for each robot; (**f**) Frame partition cost.

**Table 1 sensors-19-01994-t001:** Coverage periods for each sub-region.

Sub-region	Coverage Periods $T_{i}^{c}(s)$	Sub-region	Coverage Periods $T_{i}^{c}(s)$
1	8.00	3	40.00
2	20.00		

**Table 2 sensors-19-01994-t002:** Parameters of Algorithm 1 in Figure 13.

Parameter	Value	Parameter	Value
The number of iterations	$N_{ITE}=150$	Min speed of particle update	$V_{MIN}=-0.1$
The number of swarms	$N_{SWA}=30$	Max inertial weight	$A_{MAX}=0.9$
Particle number of each swarm	$N_{PAR}=50$	Min inertial weight	$A_{MIN}=0.4$
The number of max sensors	$S_{MAX}=160$	Acceleration constant $c_{1}$	$c_{1}=2$
Max speed of particle update	$V_{MAX}=0.1$	Acceleration constant $c_{2}$	$c_{2}=2$

**Table 3 sensors-19-01994-t003:** Parameters of Algorithm 2 in Figure 13.

Parameter	Value	Parameter	Value
The number of generations	$N_{GEN}=10000$	Population size	$N_{POP}=600$
The number of points	$N_{POI}=96$		

**Table 4 sensors-19-01994-t004:** Parameters of Algorithm 3 in Figure 13.

Parameter	Value	Parameter	Value
The number of iterations	$N_{ITE}=50$	Max inertial weight	$A_{MAX}=0.9$
The number of swarms	$N_{SWA}=500$	Min inertial weight	$A_{MIN}=0.4$
Particle number of each swarm	$N_{PAR}=1000$	Acceleration constant $c_{1}$	$c_{1}=2$
Max speed of particle update	$V_{MAX}=0.1$	Acceleration constant $c_{2}$	$c_{2}=2$
Min speed of particle update	$V_{MIN}=-0.1$		

**Table 5 sensors-19-01994-t005:** Coverage periods and actual coverage time for each route.

Route	Coverage Periods $T_{i}^{c}(s)$	Actual Coverage Time $T_{i}^{a}(s)$
Path 1	40.00	36.37
Path 2	20.00	18.81
Path 3	40.00	37.29
Path 4	8.00	6.75
Path 5	8.00	7.19

**Table 6 sensors-19-01994-t006:** Coverage periods and actual coverage time for each route.

Result	Route	Sub-region	$T_{i}^{c}(s)$	$T_{i}^{a}(s)$	Result	Route	Sub-region	$T_{i}^{c}(s)$	$T_{i}^{a}(s)$
Case 1	Path 1	2	10.00	9.33	Case 2	Path 1	2	20.00	19.47
	Path 2	2	10.00	8.68		Path 2	2	20.00	19.41
	Path 3	3	40.00	38.39		Path 3	3	30.00	28.77
	Path 4	1	12.00	10.95		Path 4	1	12.00	10.95
	Path 5	3	40.00	38.22		Path 5	3	30.00	29.03
Case 3	Path 1	2	7.00	6.30	Case 4	Path 1	2	10.00	9.33
	Path 2	2	7.00	5.74		Path 2	2	10.00	8.74
	Path 3	2	7.00	5.91		Path 3	2	10.00	6.35
	Path 4	2	7.00	5.57		Path 4	2	20.00	18.48
	Path 5	3	40.00	36.22		Path 5	3	20.00	18.83
	Path 6	1	12.00	10.95		Path 6	1	12.00	10.95
	Path 7	3	40.00	38.22		Path 7	3	20.00	18.92
						Path 8	3	20.00	18.65
Case 5	Path 1	2	10.00	6.94	Case 6	Path 1	2	10.00	9.53
	Path 2	2	10.00	8.82		Path 2	2	10.00	9.75
	Path 3	2	10.00	9.34		Path 3	2	10.00	8.97
	Path 4	3	20.00	18.28		Path 4	3	10.00	9.05
	Path 5	3	20.00	19.27		Path 5	3	10.00	9.41
	Path 6	1	8.00	7.24		Path 6	3	10.00	9.05
	Path 7	1	8.00	7.40		Path 7	1	12.00	10.95
	Path 8	3	20.00	17.20		Path 8	3	10.00	11.64
	Path 9	3	20.00	18.52		Path 9	3	10.00	10.53
						Path 10	3	10.00	10.90
						Path 11	3	10.00	10.41

**Table 7 sensors-19-01994-t007:** Parameters of Algorithm 1 in Figure 16.

Parameter	Value	Parameter	Value
The number of iterations	$N_{ITE}=200$	Min speed of particle update	$V_{MIN}=-0.1$
The number of swarms	$N_{SWA}=40$	Max inertial weight	$A_{MAX}=0.9$
Particle number of each swarm	$N_{PAR}=60$	Min inertial weight	$A_{MIN}=0.4$
The number of max sensors	$S_{MAX}=210$	Acceleration constant $c_{1}$	$c_{1}=2$
Max speed of particle update	$V_{MAX}=0.1$	Acceleration constant $c_{2}$	$c_{2}=2$

**Table 8 sensors-19-01994-t008:** Coverage periods and actual coverage time for each route.

Route	$T_{i}^{c}(s)$	$T_{i}^{a}(s)$	Route	$T_{i}^{c}(s)$	$T_{i}^{a}(s)$
Path 1	45.00	44.12	Path 5	10.00	7.86
Path 2	45.00	37.65	Path 6	10.00	8.00
Path 3	12.00	10.12	Path 7	45.00	39.14
Path 4	45.00	39.17

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
