# Peer review of "A Novel Cooperative Path Planning for Multi-robot Persistent Coverage with Obstacles and Coverage Period Constraints"

_sensors, 2019, doi:10.3390/s19091994_

Reviewer 1 Report

This paper proposed a combinatorial path planning algorithm to handle persistent coverage problem by several mobile sensors. Cooperatively coevolving particle swarm optimization (CCPSO2) and modified genetic (GA) are improved and simulations are conducted to verify the effectiveness. Several comments are listed as follows.

1. It is confused that authors stated in abstract part, "but ignored the coverage period in which points in the sub-regions are visited" in line 15. What are the "points" here? A more accurate expression is advised.

2. The contribution of this paper is not presented clearly. As stated in line 76-79,

3. Some grammar mistakes are found in this paper, such as "...optimize the number of robots to cover mission domains with a minimum quantity of robots." in line 79. Authors are suggested to modified the language of this paper.

4. Improved CCPSO2 is presented in section 2.1, however, just four heuristic statements are listed in the end without any rigorous mathematical expression. And in the simulation part, corresponding comparison should be presented to show the improvement of your heuristics.

5.  Sub-regions have different importance according to statement in this paper. It means each sub-region should be prioritized according to some criterion, "fixed coverage period" in this paper. Why fixed and how to deal with dynamic value?

6. It is suggested to give credits to several recent results on cooperative path planning, such as DOI: 10.1109/TIE.2018.2874587 and DOI: 10.1109/TSMC.2015.2500027.

7.  Two more advice for authors: 1) The algorithms proposed in this paper is offline, however, online is more valuable; 2) A real experiment is necessary to show the effectiveness of your algorithms;

Author Response

Dear Reviewer:

Thank you very much for the positive opinions on our manuscript. We think that all comments are valuable to improve our research. We have studied the comments carefully and tried our best to modify this manuscript. The revisions are delineated in the following parts. We look forward to receiving further comments from you.

Best regards

Authors

Reviewer 2 Report

The paper presents cooperative path-planning for multi-robot persistent coverage with obstacles and coverage period constraints. The paper is well written and refers most of the time appropriate papers. The introduction is satisfactory.

The authors present the results without further discussion/comparison or possibility to verify.

There has been progresses in the field, which the authors not even mention:

[1] Palacios-Gasos, Jose & Talebpour, Zeynab & Montijano, Eduardo & Sagues, C & Martinoli, A. (2017). Optimal path planning and coverage control for multi-robot persistent coverage in environments with obstacles. 1321-1327. 10.1109/ICRA.2017.7989156.

[2] Karapetyan, N., Benson, K., McKinney, C., Taslakian, P. and Rekleitis, I., 2017, September. Efficient multi-robot coverage of a known environment. In 2017 IEEE/RSJ International Conference on Intelligent Robots and Systems (IROS) (pp. 1846-1852). IEEE.

My recommendations for the authors:

Try to reduce the Section 2,3 as the main contribution is in Section 4.

Try to compare this results with [1] and [2].

Make the Matlab code also accessible for verification.

Author Response

(The authors gave the same response as above.)

Reviewer 3 Report

The paper is very well written and easy to understand. It is also very well supported with figures. The subject is not very novel, nevertheless the authors add some quite relevant contributions in the subject.

In section 5, many figures and results appear in a table form which helps compression. However, I think it would be necessary to increase the amount of text that refers to these figures and tables. The size of the tables and figures is very large. Perhaps only the most interesting cases should be shown.

I do not understand the first sentence of the conclusions (line 636). What do you mean ... "the planning results of the proposed method have global optimality"? I think these lines should be understood better.

The procedure is interesting although in future works it could be improved used for other artificial intelligence detection techniques. Some examples that must be cited are:

1.            Castaño, F., et al., Obstacle recognition based on machine learning for on-chip lidar sensors in a cyber-physical system. Sensors (Switzerland), 2017. 17(9).

2.            Castaño, F., et al., Self-tuning method for increased obstacle detection reliability based on internet of things LiDAR sensor models. Sensors (Switzerland), 2018. 18(5).

Author Response

Dear Reviewer:

Thank you very much for the positive opinions on our manuscript. We think that all comments are valuable to improve our research. We have studied the comments carefully and tried our best to modify this manuscript. The revisions are delineated in the following parts. We look forward to receiving further comments from you.

Best regards

Authors

Round  2

Reviewer 2 Report

The authors have updated the paper according to the reviewer's recommendations. The paper clarity and quality has increased.